# OpenReview forum: "Reducing Confounding Bias without Data Splitting for Causal Inference via Optimal Transport"
_ICML.cc/2025/Conference — ICML 2025 poster_

### Official Review · Reviewer_UGhX · 2025-03-13

**Overall Recommendation:** 1

**Summary:**

This paper focuses on the causal inference task, specifically in the binary and continuous treatment settings. The authors argue that data sparsity can hinder covariate distribution alignment across different treatment groups, leading to biased outcome predictions. Instead, they push all conditional marginals forward to the marginal distribution, where the formers are implemented by the generalized propensity score. Theoretically, they provide a bound for optimization. Extensive experiments are conducted to evaluate their method.

**Claims And Evidence:**

N/A

**Essential References Not Discussed:**

No

**Experimental Designs Or Analyses:**

1. In Tables 1 and 2, the large performance gaps between different baselines seem strange.
2. A visualization of the balance results for mitigating the selection bias is missing.

**Methods And Evaluation Criteria:**

Yes.

**Other Comments Or Suggestions:**

N/A.

**Other Strengths And Weaknesses:**

**Strengths:**

1. The proposed method is straightforward and easy to understand.
2. The writing in this paper is well-done, and the prerequisite knowledge is comprehensively provided.

**Weaknesses:**

1. Instead of optimizing the traditional balance term between $q_0(x)$ and $q_1(x)$, this paper enforces all conditional marginal distributions $q_t(x)$ to be aligned with the marginal distribution $q(x)$. However, $q_t(x)$ is approximated by the generalized propensity score, which is generally difficult to predict and has high variance. Thus, the effectiveness of this method is questionable.

2. Figure 1 does not highlight the unique technical contribution of this paper, such as the computation of the generalized propensity score.

3. In Tables 1 and 2, the performance gaps between different baselines are too large—they are on different scales. Additionally, the in-sample experiment is missing.

4. Why do the results of different models in Table 4 show only minor differences, while in Tables 1 and 2, the differences between models are much larger?

5. A visualization of the balance results is missing, which is crucial because it demonstrates that balance can be achieved without data splitting.

**Questions For Authors:**

See weaknesses.

**Relation To Broader Scientific Literature:**

This paper propose a new method to help address the selection bias in treatment effect estimation, which has wide applications in healthcare, ecomony, biology, and so on.

**Theoretical Claims:**

Yes, the proof for the Theorem 4.2 is correct.

---

> ### Author Rebuttal · Authors · 2025-04-01
>
> We appreciate the reviewer for the valuable comments. We will revise the submission according to the comments and responses.
>
> **Q1**
> A visualization of the balance results is missing.
>
> **A1**
> Thank you for the valuable comments.
> We have further conducted experiments to visualize the embeddings before and after representation learning based on t-SNE.
> The results shows that our method can reduce the distribution discrepancy caused by the confounding bias.
> Please kindly refer to https://anonymous.4open.science/r/ICML_figure_commit-3B64.
>
> **Q2**
> Regarding the conditional distribution estimation based on the generalized propensity score.
>
> **A2**
> 1) We exploit the generalized propensity scores to estimate the conditional distribution,
> which leverages all the training samples for distribution modeling and alignment without data splitting.
> The data splitting issue becomes even more severe in the continuous treatment setting since multiple groups are considered, and each group receives only a part of the samples,
> hampering the performance of distribution estimation and confounding bias reduction.
> Therefore, our method can achieve better performance.
> The experimental results demonstrate the effectiveness of our method.
> In addition,
> Theorem 4.3 also shows that more training samples, i.e., a large $n$, helps to achieve a better error bound.
>
> 2) Our experimental results demonstrate the effectiveness of our method involving propensity score estimation.
> In addition, existing studies have demonstrated that propensity scores are helpful for representation learning [a][b].
> We design a different approach to leverage propensity scores, i.e., to reduce the discrepancy between conditional and marginal distributions based on propensity scores.
>
> **Q3**
> The computation of the generalized propensity score is missing in Figure 1.
>
> **A3**
> Due to space limitations,
> we present the details of the computation of the generalized propensity score in Section A of the appendix.
> We will revise Figure 1 to highlight the computation of the generalized propensity score.
>
> **Q4**
> Regarding the in-sample results.
>
> **A4**
> Thank you for the valuable comments.
> We have added the results of the in-sample experiments on the binary IHDP dataset in the following table.
> Our method achieves promising performance on the in-sample setting.
>
> |          | PEHE             | MAE             | AMSE              |
> |----------|------------------|-----------------|-------------------|
> | CFR      | 1.0462 ± 1.0905  | 0.4966 ± 0.4711 | 1.0437 ± 1.0215   |
> | GANITE   | 8.0017 ± 5.3730  | 5.3945 ± 1.0848 | 13.3972 ± 10.6536 |
> | DKLite   | 5.0756 ± 6.0795  | 0.2252 ± 0.2440 | 5.5372 ± 6.1226   |
> | CausalOT | 10.2003 ± 4.5611 | 2.7824 ± 1.4760 | 8.2140 ± 9.1621   |
> | ESCFR    | 1.0019 ± 1.6507  | 0.4434 ± 0.5371 | 2.0842 ± 1.6892   |
> | ORIC     | 0.8463 ± 0.7730  | 0.3539 ± 0.3996 | 0.8173 ± 0.7391   |
>
> **Q5**
> The differences of the models are minor in Table 4 while larger in Tables 1 and 2.
>
> **A5**
> In general, the outcome values of the real-world data IHDP and News in Tables 1 and 2 are large,
> while the outcome values of the simulation data are small.
> As a result, the differences of the models on real-world data are larger compared with those on simulation data.
> Similar observations can be drawn from existing studies [c][d].
>
> [a] Counterfactual Regression with Importance Sampling Weights, IJCAI 2019.
>
> [b] Counterfactual representation learning with balancing weights, AISTATS 2021.
>
> [c] Perfect match: A simple method for learning representations for counterfactual inference with neural networks. arXiv:1810.00656
>
> [d] GANITE: Estimation of individualized treatment effects using generative adversarial nets. ICLR 2018.

---

### Official Review · Reviewer_qAwp · 2025-03-13

**Overall Recommendation:** 3

**Summary:**

This paper proposes an effective algorithm for estimating treatment effects while reducing confounding bias, applicable to both binary and continuous treatments. It employs optimal transport methods to utilize all available samples for estimating confounding bias, thereby mitigating bias and avoiding the decrease in estimation accuracy associated with splitting training samples into smaller groups for distribution alignment.

**Claims And Evidence:**

The proposed theorem on the confounding bias bound strongly supports the algorithm's design and its proof is rather valid.

Experiments compare the performance of the proposed algorithm with baseline methods in both binary and continuous treatment scenarios. Note that the algorithm's performance advantage diminishes in binary cases and in the limited continuous treatment cases shown in Appendix E. The authors may consider providing further explanations regarding this observation.

**Essential References Not Discussed:**

The related work and citations in this paper is comprehensive.

**Experimental Designs Or Analyses:**

I examined the experimental designs and analyses in Section 5 and Appendices E and F. For continuous treatments, Synthetic/IHDP experiments ran 100 trials, and News ran 20, sufficient for statistical reliability. Binary treatment experiments on IHDP (100 trials) and News (50 trials) followed similar standards. Comparisons with multiple baselines (e.g., KNN, BART, VCNet, CFRNet) are comprehensive, with appropriate metrics. Sensitivity analysis in Appendix E.2 tests hyperparameters, demonstrating robustness. The ablation study (Table 3) isolates the impact of the Wasserstein component. The design is sound, data generation protocols are clear, and no obvious methodological flaws are apparent.

**Methods And Evaluation Criteria:**

The proposed methods and evaluation criteria are well-suited to address causal inference under confounding bias. ORIC leverages optimal transport to align distributions without splitting data, overcoming the limitations of traditional methods due to reduced sample sizes—a reasonable approach since distribution estimation relies on sufficient data. The use of neural networks for representation learning and the Sinkhorn algorithm for Wasserstein distance computation is theoretically grounded.

**Other Comments Or Suggestions:**

Page 2, Line 67, and Page 6, Line 291: There is a inconsistency in the full name of the proposed algorithm ORIC.

**Other Strengths And Weaknesses:**

Strengths:
S1: The article is well-organized, and the mathematical expressions are precise.

S2: The proposed theorem on the confounding bias bound strongly supports the algorithm's design and is proven to be valid.

S3: The paper conducts experiments comparing the proposed algorithm with multiple baseline methods, demonstrating its effectiveness.

Weaknesses:
W1: The advantages of the algorithm over distribution alignment methods using different treatment groups need further comparison and explanation.

W2: The effectiveness of the algorithm concerning data splitting issues should be further demonstrated. Experimental results indicate that the performance advantage decreases in binary cases and in limited continuous treatment scenarios.

W3: The proposed algorithm introduces significant computational complexity through nested loops to calculate the optimal solution for the Wasserstein distance. The paper should clarify this point.

**Questions For Authors:**

1. I understand that the key to this paper is linking quantified confounding bias to the Wasserstein distance, which relies on several assumptions, such as the ignorability assumption for quantifying confounding bias. How does the algorithm perform when unobserved confounders exist, and the difference between $q_t$ and $q$ cannot quantify confounding bias?

2. The paper uses the Sinkhorn algorithm, which may slow down training. How does it perform on large-scale datasets?

**Relation To Broader Scientific Literature:**

This article falls within the field of causal effect estimation, particularly regarding the use of distribution alignment to reduce confounding bias.

**Theoretical Claims:**

The theoretical framework provides an upper bound on confounding error related to distribution differences characterized by optimal transport to support the algorithm's design. Additionally, experiments compare the proposed algorithm with multiple baseline methods, demonstrating its effectiveness.

---

> ### Author Rebuttal · Authors · 2025-04-01
>
> We appreciate the reviewer for the valuable comments. We will revise the submission according to the comments and responses.
>
> **Q1**
> The performance advantage of the proposed method diminishes in binary cases and limited continuous treatment cases.
>
> **A1**
> This observation is reasonable.
> Compared with the setting with more treatment values,
> for the setting of binary treatments or limited continuous treatments, more training samples fall into each group to achieve better performance of distribution modeling and alignment.
> For example, for the binary treatment setting, each group has around half of the samples,
> thus can achieve a good performance.
> However, with more number of treatments,
> each group receives fewer samples,
> which tends to decrease the performance.
>
> **Q2**
> The advantage of the algorithm over distribution alignment methods using different treatment groups.
>
> **A2**
> Distribution alignment across different treatment groups splits training data into different subpopulations,
> which cuts down the numbers of training data,
> hampering the performance of distribution modeling and confounding bias reduction.
> The issue becomes more severe in the continuous treatment setting since many different treatment values are involved,
> and each group receives only a small part of the training samples.
> Different from them, our method considers all the training samples in each treatment group,
> which leverages more training data for distribution modeling and alignment.
> The experimental results demonstrate the advantage of our method,
> and Theorem 4.3 also shows that more training data, i.e., a large $n$, helps to achieve a better error bound.
>
> **Q3**
> Regarding the computational complexity of the Wasserstein distance
> and training on large-scale data.
>
> **A3**
> The computational complexity of the Sinkhorn algorithm is in $O(n^2d)$,
> where $n$ and $d$ are the numbers of the samples and features.
>
> The following table presents the running time results.
> Our method achieves moderate time efficiency.
> For large-scale data, it is feasible to consider optimal transport on mini-batch samples, as shown in [a].
>
> 1) Continuous treatment setting on synthetic data $(\beta = 0.25)$
>
> | Methods  | Times |
> |----------|-------|
> | ORIC     | 135s  |
> | VCNet+TR | 23s   |
> | VCNet    | 17s   |
> | ADMIT    | 47s   |
> | ACFR     | 24s   |
> | DRNet    | 26s   |
> | GPS+MLP  | 25s   |
> | MLP      | 18s   |
> | GPS      | 9s    |
> | BART     | 7s    |
> | KNN      | 8s    |
>
> 2) Binary treatment setting on the IHDP-1000 data
>
> | Methods   | Times |
> |-----------|-------|
> | ORIC      | 76s   |
> | CFRNet    | 47s   |
> | DragonNet | 41s   |
> | DKLITE    | 4s    |
> | ESCFR     | 165s  |
> | CausalOT  | 4s    |
> | GANITE    | 4s    |
> | BART      | 0.2s  |
> | OLS       | 0.2s  |
> | KNN       | 0.3s  |
>
> **Q4**
> Regarding the inconsistency in the full name of the proposed method.
>
> **A4**
> We are sorry for the confusion.
> We will revise the submission accordingly.
>
> **Q5**
> Regarding the existence of unobserved confounders.
>
> **A5**
> Since we characterize the confounding bias by measuring the discrepancy between $q_t(x)$ and $q(x)$, the ignorability assumption is required. If unobserved confounders exist, the confounding bias can not be fully captured by considering $q_t(x)$ and $q(x)$ only. We will investigate the situation with unobserved confounders in the future.
>
>
> [a] Improving mini-batch optimal transport via partial transportation, ICML 2022.

---

### Official Review · Reviewer_L5M9 · 2025-03-13

**Overall Recommendation:** 3

**Summary:**

This paper proposes a novel method for causal effect estimation based on optimal transport. The method reduces the confounding bias without data splitting, which is different from the existing methods that partition training into multiple groups according to treatments. Theoretical and empirical results are provided to evaluate the performance of the proposed method. Extensive experiments on both binary and continuous treatment settings are conducted.

**Claims And Evidence:**

The claims are well supported by the theoretical analysis and extensive experiments.

**Essential References Not Discussed:**

As  far as I know, the essential references are well discussed and cited.

**Experimental Designs Or Analyses:**

The experimental designs and analyses are sound

**Methods And Evaluation Criteria:**

Benchmark datasets of both binary and continuous treatment settings are used in the experiments, and multiple evaluation metrics are adopted.

**Other Comments Or Suggestions:**

1. It seems that the analysis in Section B is related to the binary treatment setting. Is it feasible to derive similar results regarding the continuous treatment setting?

2. In Section E.1, I suggest to replace the notation $\mathcal{R}$ by $\mathbb{R}$, which is consistent with the main part of the submission.

**Other Strengths And Weaknesses:**

Strengths

1.     The paper proposes a novel method to reduce confounding bias without data splitting, which is an under-explored and interesting topic.

2.     The extensive theoretical analysis regarding confounding bias, outcome estimation error, and effect estimation error are provided.

3.     The experiments are sufficient. Both binary and continuous treatment settings are considered, multiple evaluation metrics are adopted, and many compared methods are conducted.

4.     The paper is well organized.

Weaknesses

1.     The balanced representation learning relies on the Wasserstein distance. It may bring more computation to solve the optimal transport problem.

2.     The analysis of effect estimation error in Section B only considers the binary treatment setting. Although I understand that the studies of continuous treatment usually consider potential outcome estimation instead of effect estimation, it would be better to analyze the effect estimation error of the continuous treatment setting.

**Questions For Authors:**

1. It would be better to evaluate the efficiency in terms of the running time results.

2. Is it possible to extend the proposed method to more complex settings, such as bundle or graph treatments?

3. Based on Section B, is it feasible to derive some theoretical results regarding the effect estimation error of the continuous treatment setting

**Relation To Broader Scientific Literature:**

The paper makes contributions to causal effect estimation, especially confounding bias reduction in binary and continuous treatment settings. The proposed method can be applied in different areas, such as policy decision and healthcare.

**Theoretical Claims:**

The theoretical claims are correct and well presented.

---

> ### Author Rebuttal · Authors · 2025-04-01
>
> We appreciate the reviewer for the valuable comments. We will revise the submission according to the comments and responses.
>
> **Q1**
> Regarding the effect estimation error of the continuous treatment setting.
>
> **A1**
> We analyze the effect estimation error of the continuous treatment setting below.
> Following [a], we define the effect estimation error $e_{\tau }^{G}$ in the continuous treatment setting:
> \begin{align}
> e_{\tau}^{G} = E_{t \sim p(t | t \neq 0)} E_{x \sim q(x)}[ l( h_{t}(x) - h_{0}(x), \mu_{t}(x) -\mu_{0}(x))]
> \end{align}
> With the similar procedure in Appendix B, we have:
>
> $e_{\tau}^{G}= E_{t \sim p(t | t \neq 0)} E _ {x \sim q(x)}[ l( h_{t}(x) - h_{0}(x), \mu _ {t}(x) - \mu _ {0}(x))]$
>
> $\leq E _ {t \sim p(t | t \neq 0)} [E _ {x \sim q(x)}[ l( h _ {t}(x), \mu _ {t}(x))] + E _ {x \sim q(x)} [ l( h _ {0}(x), \mu _ {0}(x))]]$
>
> $= E _ {t\sim p( t|t\neq 0)}[ \varepsilon _ {q}( h _ {t}) +\varepsilon _{q}( h _ {0})]$
>
> $= E _ {t\sim p( t)}[ \varepsilon _ {q}( h _ {t})]$
>
> $\leq \int_{\mathcal{T}} \varepsilon_{q_{t}}( h_{t}) p(t) dt+\int_{\mathcal{T}}\mathcal{W}(c,q_{t} ,q) p(t) dt.$
>
> where the first inequality holds due to the triangle inequality property, and the second inequality holds because of Eq. (12) in the paper.
>
> **Q2**
> Replace $\mathcal{R}$ with $\mathbb{R}$ in Section E.1.
>
> **A2**
> Thank you for the valuable suggestion.
> We will revise the submission accordingly.
>
> **Q3**
> Regarding the running time results.
>
> **A3**
> The following table presents the running time results.
> Our method achieves moderate time efficiency.
> 1) Continuous treatment setting on synthetic data $(\beta = 0.25)$
>
> | Methods  | Times |
> |----------|-------|
> | ORIC     | 135s  |
> | VCNet+TR | 23s   |
> | VCNet    | 17s   |
> | ADMIT    | 47s   |
> | ACFR     | 24s   |
> | DRNet    | 26s   |
> | GPS+MLP  | 25s   |
> | MLP      | 18s   |
> | GPS      | 9s    |
> | BART     | 7s    |
> | KNN      | 8s    |
>
> 2) Binary treatment setting on the IHDP-1000 data
>
> | Methods   | Times |
> |-----------|-------|
> | ORIC      | 76s   |
> | CFRNet    | 47s   |
> | DragonNet | 41s   |
> | DKLITE    | 4s    |
> | ESCFR     | 165s  |
> | CausalOT  | 4s    |
> | GANITE    | 4s    |
> | BART      | 0.2s  |
> | OLS       | 0.2s  |
> | KNN       | 0.3s  |
>
>
> **Q4**
> Is it possible to extend the proposed method to more complex settings, such as bundle or graph treatments?
>
> **A4**
> Thank you for the valuable comments.
> In general, the bundle or graph treatments remain opening problems.
> Inspired by the assumption in networked interference [b][c],
> it is feasible to aggregate the information of bundle or graph treatments into a continuous treatment value,
> so that the proposed method can be employed.
> We will investigate this challenging problem in the future.
>
> [a] Estimating heterogeneous treatment effects: Mutual information bounds and learning algorithms, ICML 2023.
>
> [b] Identification and estimation of treatment and interference effects in observational studies on networks, JASA 2021.
>
> [c] Learning individual treatment effects under heterogeneous interference in networks, TKDD 2024.

---

### Official Review · Reviewer_6NpC · 2025-03-15

**Overall Recommendation:** 3

**Summary:**

This paper extends CFRNet to use all samples for each treatment when computing loss functions. Thus, it fits the continuous treatment setting better as it does not suffer from the sample splitting problem. However, this is at the cost of modeling propensity scores and density estimation for $q(x)$ and $q_t(x)$. The theoretical results extend those in CFRNet to their setting without sample splitting. Experiments show the proposed method leads to smaller errors in PEHE, MAE and AMSE.

**Claims And Evidence:**

See below.

**Essential References Not Discussed:**

NA

**Experimental Designs Or Analyses:**

The experiments include both continuous and binary treatment.

1. For the binary treatment case, I think the proposed method is very similar to the CFRNet with Wasserstein distance, except it uses a soft propensity score to compute the error and the Wasserstein distance. I wonder how it can be much better than CFRNet in this case, especially considering the error of propensity score models and density estimation models can lead to error in counterfactual outcome prediction.

2. It would be better to add an experiment to show how the error in propensity score models and density estimation models impact the final performance of the proposed method ORIC.

**Methods And Evaluation Criteria:**

The method is an extension of the loss function proposed by CFRNet to continuous treatment.

1. The main concern of the proposed method is the computational cost of the wasserstein distance $\mathcal{W}(c_{\phi},\hat{q}_t, \hat{q})$ and the overhead of density estimation to obtain $\hat{q}_t$ and $\hat{q}$. For the Wasserstein distance, basically for each treatment $t$, each plan $\pi^t$ is an n by n kernel. I am not sure how many values of $t$ and plans are there. Can the authors discuss the time and space complexity for computing the Wasserstein distance?

**Other Comments Or Suggestions:**

NA

**Other Strengths And Weaknesses:**

NA

**Questions For Authors:**

See above.

**Relation To Broader Scientific Literature:**

NA

**Theoretical Claims:**

1. The authors claim $q_t(x) > 0$ for all $x$ given Asm 3.3, which seems not true. If $p(x)=0$, then no $q_t(x)=0$, regardless of $p(t|x)$,
2. Although it can be true intuitively, the claim that a small $\mathcal{W}(c,q_t,q)$ leads to poor outcome prediction performance due to losing information for outcome prediction is not well supported by theoretical analysis in this work.
3. Theorem 4.3 is an extension of Theorem 1 in [1] to continuous treatment, which says the outcome estimation error is upper bounded by the sum of the estimation error on factuals and the distance between the treated and controlled group in the representation space. I guess the main difference is that Theorem 4.3 allows soft $p(t)$, that is, for each $x$ you estimate the propensity score $p(t)$ and plug it into the upper bound. Could the authors clarify the difference between their contribution and existing work in terms of Theorem 4.3?

[1] Shalit, Uri, Fredrik D. Johansson, and David Sontag. "Estimating individual treatment effect: generalization bounds and algorithms." International conference on machine learning. PMLR, 2017.

---

> ### Author Rebuttal · Authors · 2025-04-01
>
> We appreciate the reviewer for the valuable comments. We will revise the submission according to the comments and responses.
>
> **Q1**
> The time and space complexity for computing the Wasserstein distance.
>
> **A1**
> 1) In practice, to avoid heavy computation, we consider a set $\widehat{\mathcal{T}}$ including sampled treatment values evenly distributed in the continuous treatment space $\mathcal{T}$. The number of treatment values is set as $[20, 100]$, as shown in Figure 3.
>
> 2) For each $t \in \widehat{\mathcal{T}}$, we apply the Sinkhorn algorithm to compute the Wasserstein distance.
> Let $n$ and $d$ be the numbers of samples and features,
> the time complexity is in $O(n^2d)$,
> and the space complexity is in $O(n^2 + nd)$.
>
> 3) We report the running time results in **A3** to Reviewer L5M9.
> Our method has moderate time efficiency.
>
> **Q2**
> Regarding $q_t(x)>0$.
>
> **A2**
> The condition $p(x) > 0$ is implicitly embedded in Asm. 3.3.
> This is because the conditional density could be rewritten as $p(t|x) = \frac{p(x,t)}{p(x)}$, where we assume $0 < p(t|x) < 1$.
> If $p(x) = 0$, then $p(t|x)$ would be undefined.
>
> **Q3**
> Regarding the claim that a small $\mathcal{W}(c,q_t,q)$ leads to poor outcome prediction performance.
>
> **A3**
> Only optimizing the Wasserstein distance without incorporating outcome prediction loss can easily lead to balanced but trivial latent representations, such as mapping all samples to a single point. This is known as the over-balancing issue, as stated in [a][b].
>
> **Q4**
> Difference between Theorem 4.3 and Theorem 1 in [c].
>
> **A4**
> 1) [c] reduces the discrepancy between the subpopulations $q_1(x)$ and $q_0(x)$, each of which is modeled by the samples in one group.
> Different from it, we reduce the discrepancy between the marginal distribution $q(x)$ and the conditional distribution $q_t(x)$,
> which is modeled by all the samples equipped with the propensity scores.
> This difference is significant since data splitting is avoided and more data are leveraged for conditional distribution modeling.
> The data splitting issue becomes even more severe in the continuous treatment setting since multiple groups are considered, and each group receives only a part of the samples,
> hampering the performance of distribution estimation and confounding bias reduction.
>
> 2) [c] applies IPM to measure the confounding bias,
> while our analysis applies the Wasserstein discrepancy to measure the confounding bias.
> Although IPM can be implemented as the Wasserstein-1 distance, the Wasserstein discrepancy based on different underlying cost functions cannot be represented as IPM.
>
> **Q5**
> Advantage of the proposed method compared with CFRNet.
>
> **A5**
> 1) CFRNet splits training data into two groups to estimate $q_1(x)$ and $q_0(x)$.
> Different from it, our method leverages all the training data to estimate $q_1(x)$ and $q_0(x)$.
> Therefore, more training data are involved in conditional distribution modeling.
> Theorem 4.3 also shows that more training samples, i.e., a large $n$, helps to achieve a better error bound.
>
> 2) CFRNet reduces the discrepancy between $q_1(x)$ and $q_0(x)$.
> Different from it,
> we aim to reduce the discrepancy between $q_1(x)$ and $q(x)$,
> and the discrepancy between $q_0(x)$ and $q(x)$,
> which can be naturally applied into the continuous treatment setting without considering pairs of different treatments and data splitting.
>
> 3) Existing studies have demonstrated that propensity scores are helpful for representation learning [d][e].
> We design a different approach to leverage propensity scores, i.e., to reduce the discrepancy between conditional and marginal distributions based on propensity scores.
>
> **Q6**
> How the error in propensity score models and density estimation models impact the final performance of the proposed method.
>
> **A6**
> Following existing works of optimal transport such as [f], we adopt $q(x)= \frac{1}{n}$ to avoid density estimation.
>
> We compare the results using the ground truth and predicted propensity scores.
> Since the ground-truth propensity score is unknown, we modified the IHDP dataset to assign treatment accordingly it.
> Our method can achieve comparable results even with the errors in propensity score models.
>
> |               | PEHE            | MAE             | AMSE            |
> |---------------|-----------------|-----------------|-----------------|
> | ground-truth ps | 1.4624 ± 0.1222 | 0.1662 ± 0.1255 | 2.1058 ± 0.1526 |
> | Estimate ps   | 1.3400 ± 0.0800 | 0.2012 ± 0.1477 | 1.9894 ± 0.1329 |
>
> [a] On learning invariant representations for domain adaptation, ICML 2019.
>
> [b] Counterfactual representation learning with balancing weights, AISTATS 2021.
>
> [c] Estimating individual treatment effect: generalization bounds and algorithms, ICML 2017.
>
> [d] CounterFactual Regression with Importance Sampling Weights, IJCAI 2019.
>
> [e] Counterfactual representation learning with balancing weights, AISTATS 2021.
>
> [f] Optimal transport for domain adaptation, TPAMI 2017.

---

### Decision · Program_Chairs · 2025-05-01

**Decision:**

Accept (poster)

**Comment:**

This paper proposes a method called ORIC for causal effect estimation of continuous or binary treatment variables. It can use all samples to compute the loss function without the sample splitting process dividing the training samples into multiple groups based on the treatment value, which is commonly relied upon by existing methods. Getting rid of sample splitting mitigate the difficulty to alignment due to insufficient sample size within a single group. Theoretically, this article gives the upper bound of the proposed method for covariate balance error and outcome estimation error. Empirically, a large number of experiments have been conducted in continuous and binary cases, verifying the superiority of the proposed method over the baselines in multiple metrics such as PEHE, MAE, and AMSE.

Overall, the reviews for this paper is positive. On the one hand, this paper is widely recognized for its good organization and writing, solid theoretical results, and extensive experimental results. On the other hand, the reviewers are generally concerned about the complexity of ORIC, and the author added the algorithm complexity analysis and experimental runtime results in the rebuttal, showing several times the time cost compared to the baselines with continuous treatment variables, and slightly smaller increase with binary variables. The author also responded to other questions from the reviewers, including questions about theoretical details, sensitivity analysis, differences and advantages compared to existing methods, experimental results in the sample, visualized experimental results, etc. Overall, considering the review comments, rebuttal, and the paper itself, I recommend that this paper be accepted.